# Effects of Early Exposure to Isoflurane on Susceptibility to Chronic Pain Are Mediated by Increased Neural Activity Due to Actions of the Mammalian Target of the Rapamycin Pathway

**DOI:** 10.3390/ijms241813760

**Published:** 2023-09-06

**Authors:** Qun Li, Reilley Paige Mathena, Fengying Li, Xinzhong Dong, Yun Guan, Cyrus David Mintz

**Affiliations:** 1Department of Anesthesiology and Critical Care Medicine, Johns Hopkins University School of Medicine, Baltimore, MD 21205, USA; r.paigemathena@gmail.com (R.P.M.); fli41@jhmi.edu (F.L.); yguan1@jhmi.edu (Y.G.); 2Solomon H. Snyder Department of Neuroscience and Center for Sensory Biology, Johns Hopkins University School of Medicine, Baltimore, MD 21205, USA; xdong2@jhmi.edu

**Keywords:** anesthesia neurotoxicity, neuropathic pain, neural activity, mammalian target of rapamycin (mTOR), dorsal spinal cord (DSC), dorsal root ganglion (DRG)

## Abstract

Patients who have undergone surgery in early life may be at elevated risk for suffering neuropathic pain in later life. The risk factors for this susceptibility are not fully understood. Here, we used a mouse chronic pain model to test the hypothesis that early exposure to the general anesthetic (GA) Isoflurane causes cellular and molecular alterations in dorsal spinal cord (DSC) and dorsal root ganglion (DRG) that produces a predisposition to neuropathic pain via an upregulation of the mammalian target of the rapamycin (mTOR) signaling pathway. Mice were exposed to isoflurane at postnatal day 7 (P7) and underwent spared nerve injury at P28 which causes chronic pain. Selected groups were treated with rapamycin, an mTOR inhibitor, for eight weeks. Behavioral tests showed that early isoflurane exposure enhanced susceptibility to chronic pain, and rapamycin treatment improved outcomes. Immunohistochemistry, Western blotting, and q-PCR indicated that isoflurane upregulated mTOR expression and neural activity in DSC and DRG. Accompanying upregulation of mTOR and rapamycin-reversible changes in chronic pain-associated markers, including N-cadherin, cAMP response element-binding protein (CREB), purinergic P2Y12 receptor, glial fibrillary acidic protein (GFAP) in DSC; and connexin 43, phospho-extracellular signal-regulated kinase (p-ERK), GFAP, Iba1 in DRG, were observed. We concluded that early GA exposure, at least with isoflurane, alters the development of pain circuits such that mice are subsequently more vulnerable to chronic neuropathic pain states.

## 1. Introduction

Modern general anesthesia successfully allows the safe performance of complex surgical and diagnostic procedures in seriously ill patients of all age groups [1]. However, clinical and preclinical studies have raised a concern that early life exposure to general anesthetic (GA) may substantially alter nervous system development resulting in functional changes, including, but not limited to, cognitive and behavioral deficits [2,3,4,5,6]. There is evidence that other aspects of neurological function, such as motor function, may be affected as well [7,8]. So, it is reasonable to consider whether development of other neuronal circuitry, such as sensory pathways mediating pain perception, could also be altered by GA exposure. 

Chronic pain is a debilitating condition that affects more than 75 million Americans and is the primary complaint for 20% of all annual outpatient clinic visits [9,10]. Pain syndromes is also a leading cause of functional disability and a major public health problem worldwide that results in a tremendous burden of human suffering [11,12]. The mechanisms underlying chronic pain are typically considered include sensitization of peripheral (PNS) and/or central nervous system (CNS) pain circuitry that is caused by impairment or inflammation [13]. The influence of early life experiences on the development of pain circuitry remains incompletely understood. It has been shown that neonatal intensive care can cause a lasting increase in susceptibility to chronic pain but given the diverse elements of this type of medical care it is not possible to isolate any single causative factor [14]. Early pain experiences or inadequate exposure to analgesics and sedatives in infants have been shown to cause long-term increases in pain sensitivity and alterations in neural development [15,16]. Furthermore, children who undergo simple and otherwise uncomplicated surgeries are commonly afflicted by post-surgical pain (CPSP) [17,18,19], which is an intermediate term pain disorder in which pain persists well beyond the period of surgical injury and healing. The risk factors for CPSP remain incompletely understood, but intriguingly may include exposure to GA [20]. Treatment options for chronic pain disorders are severely limited. Thus, if GA exposure has the potential to increase the likelihood of later-life susceptibility to chronic pain, it is of great importance to further explore and understand this phenomenon. 

The mammalian target of rapamycin (mTOR) system is a ubiquitous and complex molecular signaling pathway that integrates extracellular and intracellular sensing systems to regulate cellular metabolism and growth [21,22]. In addition, mTOR has been associated with the development and function of pain circuits [23,24,25] and inhibition of mTOR activation reduces neuropathic pain [26]. In a prior work, we found that isoflurane, one of the most commonly used GAs, aberrantly increases mTOR activity in hippocampal neurons and oligodendrocytes. This pathological alteration disrupts neuronal development and myelination in hippocampus and causes cognitive deficits [27,28]. However, the complex relationship between GA exposure, changes in mTOR signaling, and the development of pain circuitry as it relates to susceptibility to neuropathic pain remains unclear. 

Our previous work shows that, in intact newborn mice, isoflurane exposure causes cellular and molecular alteration in the developing pain perception circuitry via a pathologic upregulation of mTOR signaling pathway and enhances sensitivity to painful stimuli [29]. In this study, we sought to test the overall hypothesis that early postnatal GA exposure alters the development of pain circuitry through effects caused by increased signaling in the mTOR pathway such that mice more are rendered more susceptible to chronic neuropathic pain. To this end, we employed spared nerve injury (SNI) as a model of chronic neuropathic pain in mice exposed to GA using isoflurane. We probed for effects on pain circuitry in the dorsal spinal cord (DSC) and dorsal root ganglion (DRG), which are key areas for transmission of pain perception [30,31] that have not previously been explored in the setting of developmental effects of GAs, and we investigated the effects of anesthetics on the subsequent function of both neuronal and glial populations in those areas that are known to mediate pain perception processing [32,33]. 

## 2. Results

### 2.1. Effect of Early Isoflurane Exposure and Rapamycin Treatment on Sensitivity for Chronic Pain

Mice were exposed to isoflurane (Iso) at P7 or remained in room air as naïve or control (Ctrl). At P28, all Ctrl- and Iso-exposed mice underwent spared nerve injury (SNI). From P28 to P84, these mice were treated with rapamycin (Rapa) or vehicle (Veh) (Figure 1A). Meanwhile, three groups of animals were conducted for pain behavior tests (*n* = 12): (I) Ctrl+SNI+Veh; (ii) Iso+SNI+Veh; (iii) Iso+SNI+Rapa. All mice, except naïve, underwent SNI-evoked chronic pain. We did not set “Iso only” group in this study because effect of isoflurane exposure on intact animals had been investigated in our previous study [29].

#### 2.1.1. Mechanical Allodynia and Hyperalgesia Were Tested with von Frey Filament Application

We examined the withdrawal threshold for mechanical stimulation in the left hind paw. At baseline point (BL; P27), the average threshold for control group (1.6 ± 0.3 g) represented the normal reaction for pain in mice, and this threshold in early isoflurane exposure cases was partially lower (Iso+SNI+Veh: 1.3 ± 0.18 g; Iso+SNI+Rapa: 1.38 ± 0.25 g). At the first week after SNI, the pain threshold of all groups dramatically declined (Ctrl+SNI+Veh: 0.48 ± 0.23 g; Iso+SNI+Veh: 0.39 ± 0.16 g; Iso+SNI+Rapa: 0.36 ± 0.09 g). From week 3, the withdrawal responding to filament stimulation started to recover in Ctrl+SNI+Veh (0.72 ± 0.26 g) and Iso+SNI+Rapa (0.53 ±0.1 g), but not in Iso+SNI+Veh (0.47 ± 0.1 g). At week 5, the pain threshold in Ctrl+SNI+Veh (0.87 ± 0.31 g) continued to be elevated compared to Iso+SNI+Veh (0.53 ± 0.14 g; *p* < 0.05) and this in Iso+SNI+Rapa partially increased (0.77 ± 0.27 g; *p* > 0.05). At week 7, there were statistical differences between Ctrl+SNI+Veh (0.9 ± 0.25 g) vs. Iso+SNI+Veh (0.53 ± 0.1 g; *p* < 0.01), and Iso+SNI+Veh vs. Iso+SNI+Rapa (0.83 ± 0.21 g; *p* < 0.05) (Figure 1B). 

#### 2.1.2. Sensitivity for Thermo Perception in Chronic Pain Was Evaluated with a Hot Plate Test

At BL, the latency for nocifensive behavior in Iso groups (21.3 ± 3.7 s and 20.5 ± 2.6 s) was shorter than Ctrl group (25.2 ± 2.8 s; *p* < 0.05). At week 1, the reactive time for heat stimulation in all groups dramatically decreased (Ctrl+SNI+Veh: 7.2 ± 1.6 s; Iso+SNI+Veh: 5.3 ± 1.6 s; Iso+SNI+Rapa: 6.7 ± 1.5 s). At week 3, time taken for nocifensive behavior in all groups was slightly longer than week 1 (Ctrl+SNI+Veh: 9.0 ± 1.8 s; Iso+SNI+Veh: 6.3 ± 1.7 s; Iso+SNI+Rapa: 7.9 ± 1.3 s). At week 5, the response latency continued to increase (Ctrl+SNI+Veh: 12.9 ± 2.2 s; Iso+SNI+Veh: 8.7 ± 3.1 s; Iso+SNI+Rapa: 11.0 ± 2.3 s). The difference between Ctrl+SNI+Veh and Iso+SNI+Veh was significant (*p* < 0.01). At week 7, the heat sensation function of control animals spontaneously recovered with prolonged latency for nociceptive stimulation compared to isoflurane exposed mice (Ctrl+SNI+Veh: 14.4 ± 3.3 s vs. Iso+SNI+Veh: 9.0 s ± 3.1 s; *p* < 0.001). Meanwhile, the treatment of Rapa promoted the functional recovery in these animals (Iso+SNI+Rapa: 12.6 ±3.2 s; *p* <0.05) (Figure 1C). 

### 2.2. Effect of Early Isoflurane Exposure on mTOR Expression and Neural Activity in Dorsal Spinal Cord (DSC)

In order to evaluate the mTOR expression and neuronal activity in lumbar DSC, we conducted fluorescence immunohistochemistry (IHC) for phospho-S6 ribosomal protein (pS6) and c-fos doubled-labeling with a specific neuron marker NeuN, respectively. The pS6 is a downstream reporter for mTOR signal pathway and c-fos which is expressed in neurons following depolarization to represent neuronal activity level. We quantitatively analyzed the numbers of pS6+/NeuN+ and c-fos+/NeuN+ double labeled neurons in L4-L6 left side DSC grey matter (ipsilateral to SNI; lamina I to V) (*n* = 6). First, pS6+/NeuN+ neurons were distributed throughout dorsal horn of spinal cord, but predominantly in lamina I to III, the critical area for transmission and modulation of sensation information. Compared to naïve (22.5 ± 6.9/section), Ctrl+SNI+Veh group showed an increased number of mTOR-expressing neurons (34.5 ± 10.6/section) but the difference was not statistically significant (*p* > 0.05). The number in Iso+SNI+Veh group (68.4 ± 12.9) significantly increased than Ctrl+SNI+Veh cases (*p* < 0.001). Rapa treatment reversed this number (Iso+SNI+Rapa: 42.4 ± 14.5; *p* < 0.01) (Figure 2). 

The c-fos labeled neurons were also detected in DSC. The number of c-fos+/NeuN+ cells in Ctrl+SNI+Veh mice (16.3 ± 4.3/section) was greater than the naive (7.9 ± 3.6; *p* < 0.05) and this number was further elevated in Iso+SNI+Veh (25.4 ± 6.9; *p* <0.05). Rapa treatment (16.1 ± 5.3; *p* < 0.05) obviously attenuated this number (Figure 3). 

The effect of isoflurane exposure and Rapa treatment on activity occurred not only in neurons but also in glial cells. In naïve mice, glial fibrillary acidic protein positive (GFAP+) astrocytes were predominately observed in superficial dorsal horn (lamina I and II) (Figure 4A). Ctrl+SNI+Veh did not significantly increase the number of total GFAP+ astrocytes than naïve (61.8 ± 5.2/section vs. 50.5 ± 8.3/section; *p* > 0.05). More GFAP+ cells were seen in Iso+SNI+Veh (75.7 ± 10.1) than in Ctrl+SNI+Veh cases (*p* < 0.05). Rapa attenuated the number (57.8 ± 9; *p* < 0.01) and it was near naïve (Figure 4B). GFAP and BrdU double labeling represent astrocyte proliferation. In naïve, 55.2 ± 7.3% astrocytes were labeled with BrdU, and this ratio was increased in Ctrl+SNI+Veh (68.5 ± 7.3%; *p* < 0.05). Iso+SNI+Veh elevated the number to 83.0 ±9.9% (*p* < 0.05) and Iso+SNI+Rapa reversed it to 64.7 ± 8.1% (*p* < 0.01) (Figure 4C).

Iba1 is a specific marker for macrophage or microglia and Iba1+ cells evenly distributed in DSC. Phosphorylated P38 mitogen-activated protein kinase (p-P38 MAPK), a signaling molecule associated with microglia activation and proliferation, was expressed in Iba1+ cells (Figure 5A). In the chronic pain model, there was no significant difference for total Iba1+ cells between naïve and Ctrl+SNI+Veh mice (25.8 ± 7.8/section vs. 28.7 ± 7.9/section; *p* > 0.05). Iso+SNI+Veh (37 ± 6.1) partially increased Iba1+ number than Ctrl+SNI+Veh (*p* > 0.05), but significantly increased microglia than naïve (*p* < 0.05). Iso+SNI+Rapa (27.8 ± 5.1/section) partially attenuated the number from Iso+SNI+Veh (*p* > 0.05) and reversed the amount of Iba1+ cells near naïve (Figure 5B). The ratio of Iba1+/p-P38 MAPK+ over total Iba1+ cells represent microglial activity. The ratio in naïve and Ctrl+SNI+Veh animals was almost identical (65.2 ± 9.7% vs. 69.2 ± 8.9%; *p* > 0.05). Iso+SNI+Veh (85.8 ± 12%) increased this ratio compared to Ctrl+SNI+Veh cases (*p* < 0.05). Rapa partially reduced this ratio (73.2 ± 10.1; *p* > 0.05) than Iso+SNI+Veh and restored it near naïve (Figure 5C).

The Western blotting (WB) was performed to further verify the results from IHC experiments (*n* = 6). The ratios of pS6 intensity of immunoreactivity over β-actin in ipsilateral DSC between naïve and Ctrl+SNI+Veh were in almost same level (90.0 ± 18.5% vs. 87.0 ± 31.4%; *p* > 0.05). The ratio in Iso+SNI+Veh was dramatically higher than Ctrl+SNI+Veh group (153.0 ± 21.3%; *p* < 0.001). This number was attenuated by rapamycin treatment (105.4 ± 18.4%, *p* < 0.01) and close to naïve animals (Figure 6A). The tendency of c-fos expression in different groups was similar with pS6 study. The intensity for naïve and Ctrl+SNI+Veh groups was identical (107.7 ± 17.0% vs. 114.2 ± 17.3%; *p* > 0.05). Iso+SNI+Veh enhanced (146.8 ± 16.1%; *p* < 0.05) and Iso+SNI+Rapa reversed (113.7 ± 22.8%; *p* < 0.05) the reactive intensity (Figure 6B). We also used WB to investigate the signaling molecules N-cadherin and phosphorylated cAMP response element-binding protein (p-CREB), which are associated with excitatory synapse function and contribute to neuropathic pain in DSC. Compared to naïve (34.8 ± 9.9%), SNI-only did not (41.7 ± 21.4; *p* > 0.05), but SNI plus isoflurane did, significantly elevate (85.5 ± 17.2; *p* < 0.01) molecule level of N-cadherin. Rapamycin restored this level (69.7 ± 21.4%) close to Ctrl+SNI+Veh animals (Figure 6C). The ratio of p-CREB over GAPDH in naïve and Ctrl+SNI+Veh was almost same (52.2 ± 7.3% vs. 53.3 ± 10.6%; *p* > 0.05), but the amount of p-CREB in Iso+SNI+Veh was significantly higher (78.5 ± 15.6%; *p* < 0.01) than Ctrl+SNI+Veh. Rapa attenuated the level of p-CREB (60.7 ± 8.2%; *p* < 0.05) and this number was near naïve (Figure 6D). The astrocyte marker GFAP was examined. Compared to naïve (35.2 ± 8.4%), the ratio of GFAP band intensity over GAPDH in Ctrl+SNI+Veh was elevated (50.2 ± 7.7%; *p* < 0.05). This number was raised by SNI plus isoflurane exposure (Iso+SNI+Veh) (70.9 ± 11.1%; *p* < 0.05) and a significant recovery resulted from rapamycin treatment (50.9 ± 8.7%; *p* < 0.01) (Figure 6E). P2Y12, a G-protein-coupled purinergic receptor, is expressed in microglia and involved in pain signal transmission in DSC. The mRNA of P2Y12 was examined with quantitative real time polymerase chain reaction (qPCR) (*n* = 4). Ctrl+SNI+Veh partially enhanced the relative level of P2Y12 than naive (105.9% ± 12.6 vs. 100 ± 0.05%; *p* > 0.05) and Iso+SNI+Veh obviously increased P2Y12 (126.1 ± 9.1%) than (99.4 ± 10.1%; *p* < 0.01) (Figure 6F).

### 2.3. Effect of Early Isoflurane Exposure and Rapa Treatment on Neural Activity of DRG in Chronic Pain Model

In the present IHC study, the effect of isoflurane and Rapa treatment on mTOR expression was observed DRG, which relays nociceptive afferent information into CNS. In SNI only (Ctrl+SNI+Veh), the percentage of pS6+ neurons in DRG was not obviously higher than in naive (49.0 ± 9.4% vs. 35.8 ± 12%; *p* > 0.05). Iso+SNI+Veh increased pS6+ neurons (68.3 ± 14.8%; *p* < 0.05) and Rapa dramatically decreased this ratio (30.8 ± 10.2%; *p* < 0.001) (Figure 7A). Calcitonin-gene-related peptide (CGRP) is a pain-related neural transmitter and generally expressed in small sized (<25 µm) nociceptive neurons in DRG. In naïve and Ctrl+SNI+Veh mice, the percentage of CGRP+ neurons was almost the same (14.7 ± 5.3% vs. 17.7 ± 5.4%; *p* > 0.05). The number in Iso+SNI+Veh was increased (28.2 ± 8.2%; *p* < 0.05), and that in Iso+SNI+Rapa decreased (18.2 ± 4.3%; *p* < 0.05). In Iso+SNI+Veh group, a certain number of medium-sized (>25 µm) CGRP+ neurons were observed (Figure 7B). 

We also examined glia cells with related molecules. Connexin 43 (Cx43) is a gap junction protein expressed in satellite glial cells (SGCs) in DRG (Figure 8A). Cx43 plays an important role in neuronal coupling during neuropathic pain. Comparing to naïve (100 ± 9.8%), Ctrl+SNI+Veh did not (106 ± 13.2%; *p* > 0.05), but Iso+SNI+Veh did (128.8 ± 17.5%; *p* < 0.05), increase Cx43 intensity. Rapamycin reduced Cx43 expression (109.3 ± 12.7%; *p* >0.05) near naïve (Figure 8B). Then, we measured Iba1+ immunoreactivity of macrophage in DRG. In this chronic study, Ctrl+SNI+Veh (116 ± 27%) partially, and Iso+SNI+Veh significantly, upregulated (143.8 ± 36%) microphage. Iso+SNI+Rapa (114.3 ± 20.1%) partially reversed the effect (Figure 8C). 

In WB, the ratio of detected molecule density over standard markers, β-actin or GAPDH, was calculated. Ctrl+SNI+Veh did not significantly increase pS6 level (76.8 ± 34.9% vs. 34.7 ± 17.5%; *p* > 0.05) compared to naive. Iso+SNI+Veh elevated this number by a large margin (149.8 ± 34.1%; *p* < 0.001) and Iso+SNI+Rapa reduced the pS6 intensity (105.5 ± 13.3%; *p* < 0.05) (Figure 9A). We used phosphorylated extracellular signal-regulated kinase (p-ERK), another marker for neural activity, which is closely associated with mTOR pathway molecules and specifically induced by noxious stimulation, to detect neuronal activity in DRG. Ctrl+SNI+Veh increased p-ERK level than naïve (96.3 ± 24.5% vs. 48.8 ± 12.5%; *p* < 0.05) and Iso+SNI+Veh produced a huge increase (175 ± 42%; *p* < 0.001). Rapa reduced the p-ERK immunoreactive density (108.2 ± 29.7%; *p* < 0.01) (Figure 9B). GFAP, a molecule expressed in astrocytes in CNS and SGCs in DRG was used to evaluate the activity of SGCs here. Ctrl+SNI+Veh upregulated GFAP level than naïve (106 ± 17.3% vs. 64.8 ± 9.4%; *p* < 0.05) and Iso+SNI+Veh (148.3 ± 40.6%) further increased the intensity than Ctrl+SNI+Veh (*p* < 0.05). This number was reversed with Rapa treatment (96.3 ± 21.2%; *p* < 0.01) (Figure 9C). Similar with IHC data for Iba1, Ctrl+SNI+Veh (130.3 ± 21.4%) did not significantly increase the Iba1 level to more than the level in naïve (105.2 ± 22.7%; *p* > 0.05), and Iso+SNI+Veh (159.2 ± 22.9%) also did not enhance the intensity than Ctrl+SNI+Veh group (*p* > 0.05), However, the difference between Iso+SNI+Veh and naïve was significant (159.2 ± 22.9% vs. 105.2 ± 22.7%; *p* < 0.01). Iso+SNI+Rapa (111.7 ± 25.7%; *p* < 0.05) downregulated Iba1 reactivity from Iso+SNI+Veh (Figure 9D).

## 3. Discussion

In the present study, we report that early isoflurane exposure enhances sensitivity for mechanical and thermal stimuli in a chronic pain model caused by SNI (see Figure 1B, C). In addition, isoflurane exposure increases mTOR expression and neural activity in DSC and DRG, which are key elements of the pain-processing circuitry. The inhibition of the mTOR signal pathway with rapamycin treatment substantially ameliorates the susceptibility to pain stimuli and reverses the effects on neural activity. A range of other markers of chronic pain are upregulated by isoflurane exposure and their expression levels are reduced with rapamycin treatment. These findings suggest that early postnatal exposure to isoflurane renders mice susceptible to worsened outcomes in the SNI model of chronic neuropathic pain. The mTOR signaling pathway and related molecules in pain circuitry are involved in this alteration.

Routine GA is generally considered safe in terms of the effects on physiology and homeostasis [34,35], but there are numerous open questions about how anesthetic agents may affect brain function, particularly in vulnerable states such as the extremes of age [36]. According to prevailing theory, GA drugs act by broadly suppressing neuronal activity in the CNS by increasing inhibition of neuronal activity via agonist activity at γ-aminobutyric acid (GABA) receptors or decreasing neuronal activity by the inhibition of N-methyl-d-asparate (NMDA) receptors [37,38]. However, recent studies suggest that the unconsciousness caused by GA is mediated by more selective effects on key groups of anesthesia-activated neurons (AANs) [39]. Because GA drugs act so widely throughout the nervous system, there are a myriad of other effects that may be related to or even completely independent from their actions on consciousness. For example, GA drugs cause a profound state of analgesia that can be shown to be independent of actions on consciousness [40], and it is unclear how activation of these neurons during the early postnatal period may impact the development of pain perception circuitry. The actions of GA drugs are not limited to the brain but can also be observed in the spinal cord and DRG neurons [41,42,43]. The DSC is a critical region for processing and transmission of neural signaling related to sensation perception [30,44], and the projection neurons in DSC relay primary nociceptive information from peripheral receptors to pain circuitry in the brain stem and cortex. Glial cells interact with nociceptive neurons in DSC by secreting neuroactive signaling molecules that modulate pain signals as they ascend in the processing system [32]. The DRG is effectively a functional gateway for primary sensation, diversely ranging from touch and proprioception to normal and pathologic pain modalities, including neuropathic pain. Neurons in DRG serve primarily to relay sensation information, and non-neuronal cells, mainly macrophages and satellite glial cells (SGCs), modulate the sensory information via neuron–glia interaction and cytokine release [45,46]. Our data show that, in a chronic pain model, early isoflurane exposure leads to a chronic increase in neuronal and associated glial activity in DSC and DRG, which effectively alters the baseline tone of the system such that there is a higher predisposition to chronic pain.

The mTOR protein is an obligatory component of two complexes (mTORC1 and mTORC2) which are at the center of a signaling system that regulates cellular activities including proliferation, differentiation, apoptosis, metabolism, transmitter release, synaptic formation, and other biological processes [21]. In naive rodents, mTOR and downstream molecules are engaged in baseline levels of signaling in key elements of pain processing circuitry at relatively low levels [47], but in a variety of pathological conditions, such as injury or inflammation, the activity of mTOR pathway molecules is dramatically increased [48,49]. Considering the correlation between enhancement of mTOR in pain circuitry and generation of pain, our findings suggest a possible mechanism by which early exposure to isoflurane contributes to chronic pain via mTOR pathway. Our finding that pharmacologic restoration of a lower mTOR activity level and resultant reductions in activity in neuronal and glial populations that engage in pain processing suggests mTOR is a key mediator of GA-induced enhanced sensitivity to chronic pain (see Figure 2, Figure 3 and Figure 6A,B). However, it should be noted that Rapa could be acting via off target effects or that the effects we observed in DSC and DRG might be indirectly caused by mTOR, which could be acting primarily through brain pain perception areas. Also, our data did not really distinguish between effects mediated by mTORC1 and mTORC2, as the rapamycin paradigm used here was expected to inhibit both pathways [50], and previous work on this topic suggests that isoflurane acts simultaneously to upregulate both of these pathways [51]. Despite these limitations, our findings are highly suggestive of mTOR-mediated mechanisms of chronic pain caused by early GA effects on DSC and DRG circuitry.

We also examined the effects of isoflurane on molecules known to play key roles in pain perception transduction and processing in DSC and DRG. N-cadherin is a trans-synaptic cell adhesion molecule expressed at glutamatergic synapses that mediates synaptic plasticity generally and which has been shown to serve this function in pain circuitry. CREB, which can be activated by N-cadherin, is involved in pain modulation in the spinal cord during the transition from acute to chronic pain [52]. An in vitro study indicated N-cadherin/CREB signaling molecules are regulated with mTOR activity [53]. In the present study, we found that early exposure to isoflurane upregulated expression of N-cadherin and CREB in the setting of SNI, whereas SNI alone had a much less obvious effect in this chronic model. We found that Rapa treatment which alleviated isoflurane-induced chronic pain in this model also reduced levels of CREB phosphorylation, but not expression levels of N-Cadherin (see Figure 6C,D). Thus, while it appears that the N-Cadherin/CREB pathway was involved, it is unclear whether the effects were mediated by changes in expression or activity of molecules in this pathway and further exploration would be required to clarify this distinction. 

In CNS and PNS, glial cells also play an important role in pathological pain. Here, we investigated three kinds of them: astrocyte, microglia/macrophage, and SGC. Astrocytes contain GFAP as the specific marker and perform numerous critical functions such regulation of extracellular ion concentration and modulation of synaptic transmission. In neuropathic conditions, astrocytes lose their ability to maintain the homeostatic concentrations of extracellular potassium (K+) and glutamate, leading to neuronal hyperexcitability [32]. Microglia serves as the resident immune cells in CNS and produce inflammatory mediators in response to pathological events including neuropathic pain. p38 MAPK, a type of protein kinase, expressed in Iba1+ glia and functions activation of microglia [54,55]. P2Y12 is a subtype of the purinergic chemoreceptor which contributes to microgliosis [56]. The functions of p38 MAPK and P2Y12 interact and both of them are expressed in microglial cells in DSC [57]. In the present study, the molecules related to astrocyte and microglia activity were upregulated in DSC in isoflurane-exposed mice with chronic pain (see Figure 4, Figure 5 and Figure 6E,F). Our data correspond with a previous observation that the mTOR pathway is involved in the expression of GFAP, p38 MAPK, P2Y12, and the activation of astrocyte and microglial cells in the chronic pain model [49,57,58]. 

The level of mTOR and a related molecule p-ERK in DRG neurons is low in normal condition, and it is activated with nerve injury [47,48,59]. Our data indicate that early GA aggravates the expression of mTOR and p-ERK that accompanies increasing pain. Rapa reverses these effects (see Figure 7A and Figure 9A,B). CGRP is a proinflammatory cytokines existing in nociceptive neurons (usually small cells). The production of this polypeptide in neuropathic pain has a close relationship with mTOR pathway [60,61]. Our observations correspond with these studies (see Figure 7B).

Like their CNS counterparts, GFAP-positive satellite glial cells (SGCs) contribute to chronic pathological pain in DRG. SGCs surround the somata of neurons and directly coupled the neurons via Cx43-containing gap junctions [62]. When nociceptive single is transported to DRG, the activity in neurons will be amplified by the “neuronal coupling” via SGCs [63,64]. Our data showed that early GA exposure increases Cx43 and in SGCs and GFAP in DRG. This alteration is mTOR-dependent (see Figure 8A,B and Figure 9C). Paralleling the activation of Iba1+ microglia in DSC with neuropathic pain is a significant expansion and proliferation of Iba1+ macrophages around DRG neurons [65]. The data in the present study indicate GA upregulates Iba1+ macrophages and Rapa treatment reverses this alteration (see Figure 8C and Figure 9D). An upregulation in inflammatory signal may either be causative or complementary to the observed changes in neuronal activity.

In the present study, both SNI and isoflurane exposure caused an increase in neural activity and mTOR molecule level. Although previous acute or sub-acute (<14 days) studies showed that SNI elevates neuronal activity and mTOR protein level in pain circuitry [26,48,60,66], in our chronic experiments (8 weeks after injury), the effect of SNI on cellular and molecular alteration became almost not significant. However, early isoflurane exposure to mice with SNI produced an obvious enhancement for expression of mTOR and other related molecules in the pain circuitry, and this effect lasted for a long period. These cellular and molecular data indicated that early GA exposure is a more important factor for inducing neuronal activity and causing chronic neuropathic pain. In the SNI-induced chronic pain model, the allodynia can be gradually recovered, but this recovery is very difficult in cases with both early GA exposure cases. 

In a chronic pain model, early isoflurane exposure enhances sensitivity for nociceptive stimulation. Meanwhile, isoflurane aberrantly increased the neural activity and enhanced the mTOR signaling pathway in CNS and PNS circuits related to nociceptive sensation. The impediment of mTOR activity attenuated these alterations. We did not test other GAs in this investigation, and while it is likely that our findings will generalize at least to other halogenated ether GAs and perhaps to chemically different intravenous agents that also act on GABA receptors, this hypothesis remains to be substantiated. We believe the significance of our findings is that it should prompt research in two further directions: (1) It would be very instructive to conduct prospective studies in patients undergoing early life anesthesia to test whether modifications in anesthetic technique and/or preventative strategies to mitigate the hypothesized anesthetic effect can affect chronic pain outcomes. (2) We believe that further mechanistic investigation of the anesthetic effects on the development of pain circuitry are independently interesting in uncovering potential mechanisms to prevent chronic pain, contributing to our understanding of the development of pain circuitry and its relationship with chronic pain, and also to better describe the putative effects of anesthetics on the developing nervous system.

## 4. Materials and Methods

### 4.1. Animal Paradigm and Experimental Timeline

A total of 64 immature C57BL/6 mice were used in this study. The body weight was in the range of 4.1–4.5 g at postnatal day 7 (P7). Both sexes were equally represented in all experiments. All study protocols involving mice were approved by the Animal Care and Use Committee at Johns Hopkins University (Baltimore, Maryland) under approval number MO23M213 and conducted in accordance with National Institutes of Health (Bethesda, MD, USA) guidelines for care and use of animals. 

Thirty-two animals were exposed to isoflurane (Iso) for 4 h at P7 and another thirty-two mice remained in room air as naïve or control (Ctrl). At P28, sixteen unexposed (Ctrl) and all isoflurane-exposed mice underwent spared nerve injury (SNI). From P28 to end of experiment (P84), these mice were intraperitoneally injected with rapamycin (Rapa) or vehicle (Veh) twice a week. Hence, total animals were divided into four cohorts (*n* = 16): (i) Naïve; (ii) Ctrl+SNI+Veh; (iii) Iso+SNI+Veh; and (iv) Iso+SNI+Rapa. Meanwhile, from P28 to P84, animals were biweekly examined with pain behavior tests (von Frey and Hot Plate). At P84, animals were sacrificed and DSC and DRG tissue was studied with IHC (*n* = 6), Western blotting (*n* = 6), and q-PCR (*n* = 4) (Figure 1A). 

### 4.2. Isoflurane Exposure 

At P7, mice which were selected as “Iso” groups underwent isoflurane exposure. The others stayed in room air as a naïve or control (*n* = 16 for each). Volatile anesthesia exposure was accomplished using a Supera (USA) tabletop portable nonrebreathing anesthesia machine (Supera. Clackamas, OR, USA). A total of 3% isoflurane mixed in compressed air (21% oxygen and 79% nitrogen) was initially delivered in a closed chamber with selected mice for 3 to 5min. After loss of the lighting reflex, animals were transferred to specially designed plastic tubes. A heating pad (37 °C) was placed underneath the exposure setup. The mice were exposed to 1.5% isoflurane carried in the air for 4 h. A calibrated flowmeter was used to deliver carry gas at a flow rate of 5 L/min, and an agent-specific vaporizer was used to deliver isoflurane. After the isoflurane exposure, the mice were placed in room air for 2–3 min until normal respiration was observed. Then, these pups were returned to their mothers together with their littermates upon regaining righting reflex. Experimental procedures followed the modified protocols from a previously published article [29]. 

### 4.3. Spared Nerve Injury (SNI)

At P28, Animals were anesthetized by intraperitoneal injection of with ketamine and medetomidine cocktail (75/0.5 mg/kg BW) in saline. Artificial tear jelly was applied to the eyes. The mice were placed on the heating pad (37 °C) and kept on their left lateral side. The hair in left thigh was shaved off and the skin was cleaned with povidone iodine liquid. The surgical operation commenced when reflexes to painful stimuli were stopped, and procedures were manipulated under the stereo microscope. A clear depression was visible between the anterior and posterior muscle groups. A 0.8 cm long skin incision was made along this depression from the fibular head directed vertically downward. The common peroneal nerve was visible under the posterior group of muscles running almost transversely. After the muscle was raised with fine-tipped forceps and then incised with a pair of micro scissors, the common peroneal nerve was clearly exposed. The nerve trunk was isolated from surrounding fascia using fine tweezers and then was slowly ligated with gut suture 6-0. Usually, muscle contraction of the limb was observed as a tight surgical knot was made. The nerve was cut below the knot with micro scissors and a proximal end with knot was embedded underneath muscle. The muscle layer was closed with gut suture 6-0 and skin was seamed with silk suture 5-0. The wound was wiped by cotton-tip with 75% ethanol and a drop of 2% lidocaine was put on it. 0.01 mL atipamezole (Antisedan, Pfizer, New York, NY, USA) was intramuscularly (i.m.) injected to wake up animals from anesthesia. All mice were post-surgically kept in a 37 °C warming chamber for at least 1 h until they were totally woken up and the normal movement was observed. 

### 4.4. Rapamycin (Rapa) Injection 

From postnatal day 28 (following SNI) to postnatal day 84, mice received Rapa or vehicle injection twice a week. Then, 100 µL 0.2% Rapa (Sigma-Aldrich, St. Louis, MO, USA) dissolved in vehicle solution was intraperitoneally injected to animals of Iso+SNI+Rapa. The same value of vehicle was injected to Ctrl+SNI+Veh and Iso+SNI+Veh groups. Vehicle solution consisted of 5% Tween 80 (Sigma-Aldrich), 10% polyethylene glycol 400 (Sigma-Aldrich), and 8% ethanol in saline. No treatment was applied for naive animals. 

### 4.5. Behavior Tests 

Animals received pain behavior tests. The von Frey test and hot plate test were performed at following time points: BL (baseline; 1 day before SNI), 1 week, 3 weeks, 5 weeks, and 7 weeks after SNI. Comparison was conducted among three groups (*n* = 12): (i) Ctrl+SNI+Veh; (ii) Iso+SNI+Veh; and (iii) Iso+SNI+Rapa.

Von Frey test. Mechanical hyperalgesia in the mice was determined by applying von Frey filaments (Smith and Nephew Inc., Germantown, WI, USA) to the left plantar surface of the hind-paws. Before being tested, animals were acclimatized for 60 min to the testing environment, which featured individual plexiglas cubicles over a coated wire mesh platform. A series of calibrated von Frey filaments (0.04 g; 0.16 g; 0.4 g; 0.6 g; 1.0 g; 1.4 g; 2.0 g) were applied for 5 s with enough force to cause filament buckling, and withdrawal response of the hind-paw was observed. The smallest filament to evoke ≥3 withdrawal responses out of five repeated applications was determined, and the average value (gram) from each hind paw was recorded as the withdrawal threshold.

Hot plate test. The conventional hot plate test was used to determine the heat thresholds of the left hind paw. Unrestrained mice were placed on a metal surface maintained at a constant temperature of 54 °C. The response latency, which was the time taken to observe a nocifensive behavior, was recorded by the investigators. Nocifensive behaviors include left hind paw withdrawal or licking, stamping of the leg, leaning posture, or jumping. If no nocifensive behaviors were observed in 30 s (cut-off time), animals were removed from the hot plate to prevent tissue damage. Each animal was tested 3 times with at least a 30 min time interval. The average value was recorded as final time latency.

### 4.6. Immunohistochemistry (IHC) 

During P81 to P84, 5-bromo-2′-deoxyuridine (BrdU; Abcam, UK) was injected intraperitoneally at 50 mg/kg daily in animals for IHC study (*n* = 6 per group). At P84, these mice were perfused with 30 mL 0.1M phosphate-buffered saline (PBS) followed by 40 mL 4% paraformaldehyde in PBS. L4–L6 lumbar spinal cords which correspond to the levels for sciatic nerve, and left L4-L6 DRGs were removed, post-fixed overnight, and cryo-protected in 30% sucrose for 48 h. The spinal cords were transversally sectioned in 20 µm and DRGs were cut in 10 µm with a cryostat and sections were directly mounted on glass slides in a rotating order. The spinal cord and DRG sections on slides were stained using immunostaining chambers (Thermo Scientific, Waltham, MA, USA). For BrdU staining, spinal cord sections were pretreated with 2N HCl to denature DNA (37 °C; 45 min), and with 2 × 15 min borate buffer (pH 8.5) to neutralize the HCl. Then, all sections were blocked in 10% normal goat serum (Sigma Aldrich, St. Louis, MO, USA) and 0.1% triton X-100 for 60 min, followed by primary antibody incubation at 4 °C overnight. Primary and secondary antibodies were purchased from reliable companies according to published articles. Before IHC, negative control (omit primary antibody) and positive control experiment were performed to confirm antibody specificity. For positive control, immunolabeled cells were same as previously published images. No immunoreactivity was observed in negative control. The following primary antibodies were used for double or single immunolabeling: (1) rabbit anti-phospho-S6 (pS6) (1:1000; Cell Signaling, Danvers, MA, USA; Cat#: 2211); (2) mouse anti-c-fos (1:1000; Abcam; Cambridge, MA, USA; Cat#: ab298942); (3) mouse anti-NeuN (1: 200; EDM Millipore; Darmstadt, Germany; Cat#: MAB377); (4) rabbit anti-NeuN (1:200; EDM Millipore; Darmstadt, Germany; Cat#: ABN78); (5) mouse anti GFAP (1:1000; Millipore-Sigma; Burlington, MA, USA; Cat#: MAB360); (6) rabbit anti-BrdU (1:500; Rockland; Limerick, PA, USA; Cat#: 660-410-C29); (7) rabbit anit-Iba1 (1:1000; Wako; Osaka, Japan; Cat#: 019-19741); (8) mouse anti-p-P38 MAPK (1:200; Cell Signaling, Danvers, MA, USA; Cat#: #9216); (9) Mouse anti-CGRP (1:1;000; Abcam; Cambridge, MA, USA; Cat#: ab81887); (10) Rabbit anti-connexin-43 (Cx43) (1:500; Sigma-Aldrich; St. Louis, MO, USA; Cat#: C6219). After 3 × 10 min washes in PBS, sections were incubated with secondary antibodies: Cy3 conjugated goat anti-rabbit (Cat#: 111-165-144) or goat anti mouse (Cat#: 115-165-146) IgG (1:600; Jackson Laboratory, West Grove, PA USA) mixed with Alexa 488 conjugated goat anti-mouse (Cat#: A11029) or goat anti-rabbit (Cat#: A11034) IgG (1:300; Invitrogen, Eugene, OR, USA) for 2 h. After 3 × 10 min PBS washes, slides with DSC or DRG sections were separated from chambers. All slides were air-dried and cover-slipped with Prolong Gold antifade to allow for imaging without substantial loss of signal (Invitrogen, Eugene, OR, USA; Cat#: P36930). 

### 4.7. Imaging Analysis for IHC

Immunostained sections were observed and photos were taken with a Leica 4000 confocal microscope (Leica, Wetzlar, Germany). Images were quantitatively analyzed using the ImageJ program (NIH, Bethesda, MD, USA). All immunolabeled cells, including single channel identified and double-labeled cells in channel-merged images, were counted with “cell counter plugin” > “cell-counting” tool of ImageJ. The total numbers for each section were recorded. In left DSC, gray matter (lamina I-V) region was outlined. All pS6+/NeuN+ and c-fos+/NeuN+ cells in this area were counted to show mTOR in neurons and neuronal activity. BrdU+/GFAP+ cells represent proliferating astrocytes and the ratio of these cells over total GFAP+ means activity status of astrocyte. Iba1+ cells were considered to be microglia and p-P38 MAPK, a signal pathway contributing to microglia activation, was exclusively expressed in Iba1+ cells. The ratio of p-P38 MAPK+ over all Iba1+ cells indicate the level of microglia activation. In DRG, the percentage of pS6+ neurons over total neurons (pS6+ plus dim background cells), and CGRP+ over total NeuN+ neurons, were calculated. Cx43 (a gap junction protein)-labeled satellite glial cells (SGCs) and Iba1-positive microglia-like cells were investigated in DRG by measuring cellular fluorescence intensity of these two markers in each section using ImageJ with “integrated density” tool. The ratio of average intensity from each group over naïve was calculated. The diameter of CGRP neurons were measured with ImageJ according to following criteria: diameter less than 25 µm was considered as small cell; 25–50 μm as medium size; and >50 μm as large cell. In this study, identical photo exposure was set for all groups. Two researchers participated in the experiments. The person performing image analysis was blinded to the experimental grouping in all cases.

### 4.8. Western Blotting (WB) 

Animals (*n* = 6 per group) were quickly perfused with cold saline at P84. L4-6 dorsal spinal cords and the left L4-6 DRGs were immediately removed, and quickly frozen in dry ice. The frozen spinal cord was first separated into left and right parts with a blade along midline and then into ventral and dorsal parts using a micro scissors in dissection microscope. The tissue of left DSC and DRG was lysed in the lysis buffer (10% glycerol, 0.1% NP40, 0.5% phosstop, and 0.5% cOmplete in TBS) and homogenized with a bullet bender (Next Advance, Troy, NY, USA). After centrifuging, the supernatant was taken. Samples were prepared with 1:1 denaturing sample buffer (Bio-Rad, Hercules, CA, USA), boiled for 5 min, and run on Novex 4–12% Bis-Tris Protein Gels (Thermo Fisher, Waltham, MA, USA) in NuPage running buffer (Thermo Fisher, Waltham, MA, USA) with 150 volts for about 1 h. The proteins were transferred to Novex nitrocellulose-blotting membranes (Thermo Fisher, Waltham, MA, USA). Blots were incubated in TBST buffer with 5% powder milk for 1 h, and then probed in 4 °C overnight with the following primary antibodies in same buffer: rabbit anti-pS6 (1:1000); rabbit anti-c-fos (1:1000); rabbit anti-N-cadherin (1:1000); rabbit anti-phosphorylated cAMP respond element-binging protein (p-CREB) (1:1000); rabbit anti-GFAP (1:10; ImmunoStar); mouse anti phosphorylated extracellular signal-regulated kinase (p-ERK) (1:200; Santa Cruz); anti-Iba1 (1:1000; Wako); rabbit anti-glyceraldehyde-3-phosphate dehydrogenase (GAPDH) (1:10,000); and rabbit anti-β-actin (1:2500). All primary antibodies, except these mentioned above, were purchased from Cell Signaling Technology (Danvers, MA, USA). The wet blots were incubated in HRP-linked anti-rabbit or anti-mouse IgG (Cell Signaling Technology, Danvers, MA, USA) in 5% milk for 1 h and visualized using Pierce Biotechnology ECL Western blotting substrate kit (Thermo Fisher, Waltham, MA, USA). Images were acquired using ChemiDoc imaging system (BioRad, Hercules, CA, USA). Quantitative analysis was performed with the ImageJ program (NIH, v.1.52p, Bethesda, MD, USA). The ratios of band density of pS6, c-fos, N-cadherin, and Iba1 over β-actin; or GFAP, p-CREB, and p-ERK over GAPDH; were, respectively, calculated. 

### 4.9. Quantitative Real-Time PCR (qPCR) 

Mice for qPCR analysis (*n* = 4 for each group) were sacrificed with ketamine/medetomidine anesthesia at P84, and the left L4-6 dorsal spinal cords were removed. The tissue was immediately frozen with dry ice and then placed into −80 °C for storage. The total RNA samples were prepared with “RNeasy Plus Micro Kit” (Qiagen; Cat# 74034). The cDNA synthesis reaction was performed with 1 μg of total RNA plus “qScript cDNA SuperMix” (Quanta Biosciences; Cat# 95048). The specific rat P2Y12 primers (F: 5′-GATTGATAACCATTGACC-3′; R: 5′-GGTGAGAATCATGTTAGG-3′) was applied [67]. The qPCR) was performed using Bio-Rad “SsoAdvanced Universal SYBR Green Supermix” (Cat# 172-5271) in Bio-Rad CFX96 Real-Time System (C1000 Tauch Thermal Cycler). The annealing temperature was set at 53 °C and the number of cycles was fixed to 40. Gene expression was analyzed using the ΔΔCT method with CT as the threshold cycle. The relative levels of target genes, which were normalized to naive CT value, were reported as 2^−ΔΔCT^.

### 4.10. Statistical Analysis

Statistical comparisons were performed with the Prism (GraphPad, v.8, La Jolla, CA, USA) program. Data of pain behavior tests were conducted by two-way ANOVA. Other data from IHC, WB, and q-PCR were generated by ordinary one-way ANOVA. A post hoc Tukey test was used for inter-group comparisons among groups. Normal distribution was verified prior to ANOVA testing. The criteria for significant difference were set a priori at *p* < 0.05. Results were expressed as mean ± standard deviation (SD). 

## 5. Conclusions

In summary, we conclude that early GA exposure alters the development of pain circuits and has the potential to contribute to chronic neuropathic pain and/or other pain syndromes, which is presumably the result of many factors related to disease and medical care. Meanwhile, the signal-pathway-dependent enhancement of neural activity and alteration of pain related molecules occur in DSC and DRG. These data suggest very interesting future directions to investigate potential mechanisms by which isoflurane and other anesthetics could increase the risk of chronic pain in children who undergo GA exposure.

## Figures and Tables

**Figure 1 ijms-24-13760-f001:**
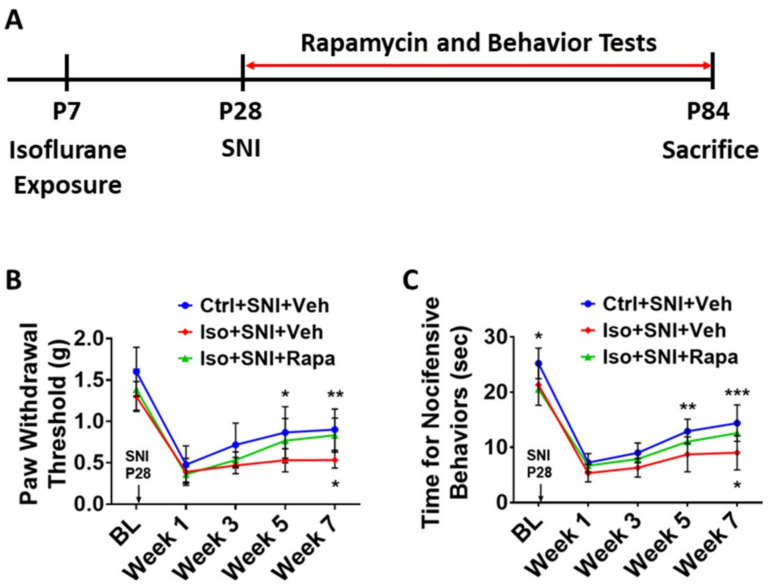
Experimental timeline and pain behavior tests. (**A**) At postnatal day 7 (P7), animals were exposed to isoflurane (Iso) for 4 h or remained in room air as naive and control (Ctrl). At P28, all Ctrl- and Iso-exposed mice underwent spared nerve injury (SNI). From P28 to P84, these mice were treated with mTOR inhibitor rapamycin (Rapa) or vehicle (Veh) twice a week. Meanwhile, animals were biweekly examined with pain behavior tests. At the end of the experiment (P84), animals were sacrificed for immunohistochemistry, Western blotting, and q-PCR analysis. (**B**) von Frey test (*n* = 12). Mice were tested at five time points: baseline (BL), week 1, week 3, week 5, and week 7 after SNI. First, SNI decreased hind paw withdrawal threshold for filament stimulation in all groups. Overall, threshold in Iso+SNI+Veh (red) mice was lower than Ctrl+SNI+Veh (blue). Rapa injection (Iso+SNI+Rapa) partially restored the threshold (green). (**C**) Hot plate test (*n* = 12). The nocifensive latency period to heat stimulation was recorded at five time points mentioned above. At BL, the latency time in Iso-exposed animals was shorter than Ctrl group. After 1 week following SNI, the latency for all groups started to recover with different degrees. Mice in Iso+SNI+Veh (red) took shorter time for nocifensive behaviors than these in Ctrl+SNI+Veh (blue) and Iso+SNI+Rapa (green). Two-way ANOVA was conducted for B and C. The asterisks above the curves indicate the difference between Ctrl+SNI+Veh and Iso+SNI+Veh; these below curves represent difference between Iso+SNI+Veh and Iso+SNI+Rapa. *: *p* < 0.05; **: *p* < 0.01; ***: *p* < 0.001. Error bars: standard deviation (SD).

**Figure 2 ijms-24-13760-f002:**
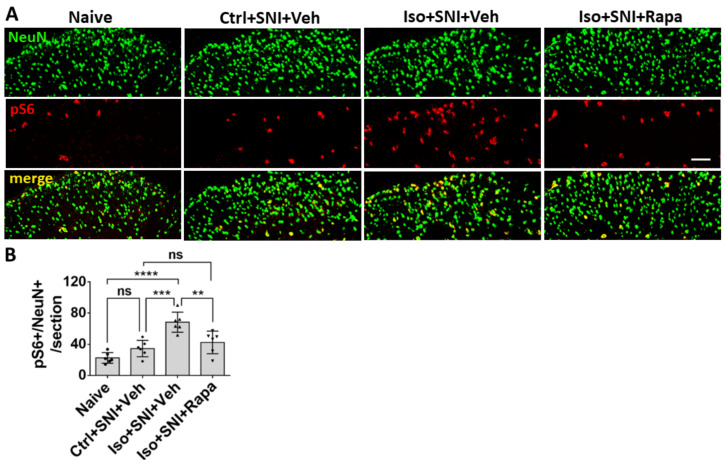
Effect of isoflurane exposure on neuronal activity in superficial DSC. (**A**) The photographs show pS6 (red channel) double-immunolabeled with NeuN (green) in DSC neurons (lamina I-II) in each group. (**B**) The histogram showed statistical results (*n* = 6). Compared to naïve group, Ctrl+SNI+Veh group showed a trend of increased number of pS6 positive neurons. Iso+SNI+Veh group showed significantly increased pS6+/NeuN+ cells, as compared to Ctrl+SNI+Veh group and naïve group. Rapa treatment reduced this change. Bar = 50 μm. One-way ANOVA for statistics. ns: no significance; **: *p* < 0.01; ***: *p* < 0.001; ****: 0.0001.

**Figure 3 ijms-24-13760-f003:**
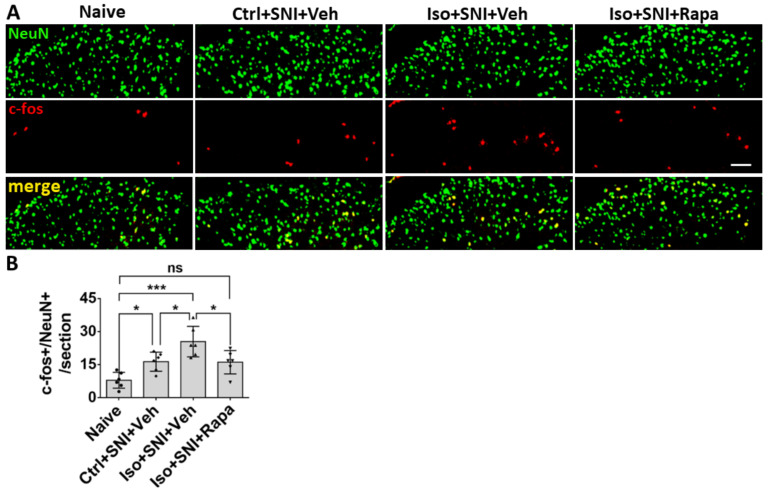
The c-fos labeled neurons in superficial DSC. (**A**) The number of c-fos+/NeuN+ cells in Ctrl+SNI+Veh mice was greater than the naive and this number was further elevated by Iso+SNI+Veh. Rapa treatment (Iso+SNI+Rapa) restored this number near the naive. (**B**) Graphs showed quantitative data (*n* = 6). Bar = 50 μm. One-way ANOVA for statistics. ns: no significance; *: *p* < 0.05; ***: *p* < 0.001.

**Figure 4 ijms-24-13760-f004:**
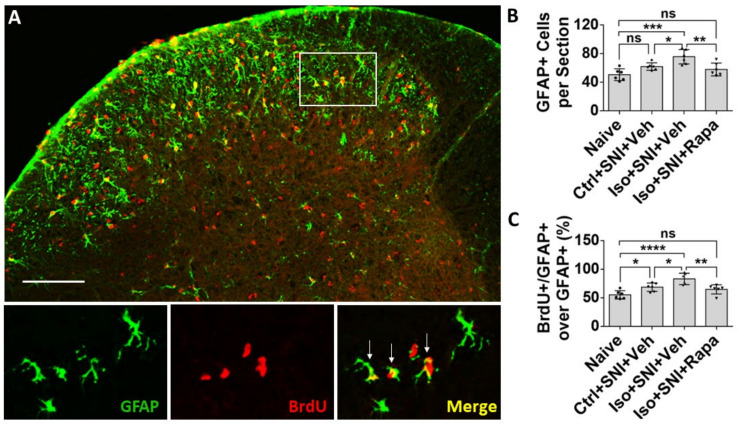
Effect of isoflurane exposure on astrocyte activity in DSC. (**A**) A micrograph with GFAP labeled astrocytes and BrdU-positive dividing cells in DSC. Astrocytes are predominately distributed in superficial dorsal horn (lamina I and II), and BrdU+ cells evenly in whole DSC. The box represented the location of high-power images below. Arrows indicated GFAP+/BrdU+ proliferating astrocytes. (**B**) The number of total GFAP+ cells in Ctrl+SNI+Veh was partially increased than naïve and more astrocytes were seen in Iso+SNI+Veh cases. Rapa restored the number near naïve (*n* = 6). (**C**) Quantitative results for GFAP+/BrdU+ proliferating astrocytes (*n* = 6). The ratio of GFAP+/BrdU+ over total GFAP+ cells in Iso+SNI+Veh was higher than naïve and Ctrl+SNI+Veh. Rapa decreased the ratio near naïve. Bar = 100 μm. One-way ANOVA. ns: no significance; *: *p* < 0.05; **: *p* < 0.01; ***: *p* < 0.001; ****: *p* < 0.0001.

**Figure 5 ijms-24-13760-f005:**
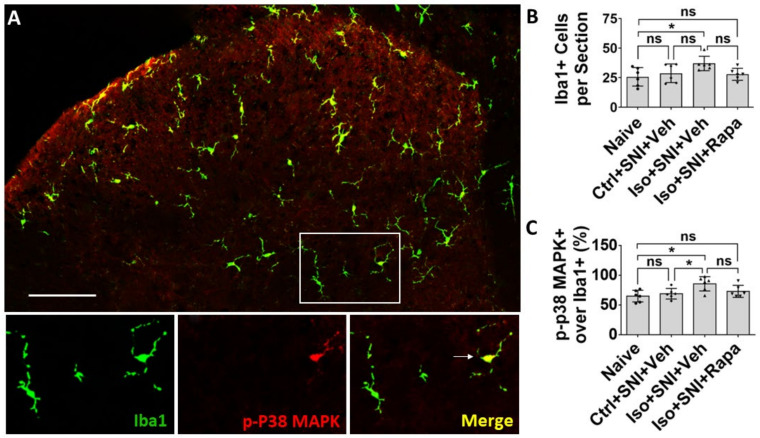
Effect of isoflurane exposure on microglia activity in DSC. (**A**) Iba1 positive microglia evenly distributed in DSC and p-P38 MAPK was expressed in Iba1+ cells. The box showed the location of high-power images below. Arrow indicated an Iba1+/ P38 MAPK+ microglia. (**B**) Quantitative results for total microglia numbers (*n* = 6). In this chronic pain model, there was no difference for Iba1+ cells between naïve and Ctrl+SNI+Veh mice. Iso+SNI+Veh increased Iba1+ number than naive. Iso+SNI+Rapa partially attenuated this number near naïve. (**C**) Quantitative data for microglial activity (*n* = 6). The percentage of Iba1+/p-P38 MAPK+ over total Iba1+ cells in naïve and Ctrl+SNI+Veh animals was identical. The number in Iso+SNI+Veh was increased and Iso+SNI+Rapa partially reduced this ratio near naïve. Bar = 100 μm. One-way ANOVA. ns: no significance; *: *p* < 0.05.

**Figure 6 ijms-24-13760-f006:**
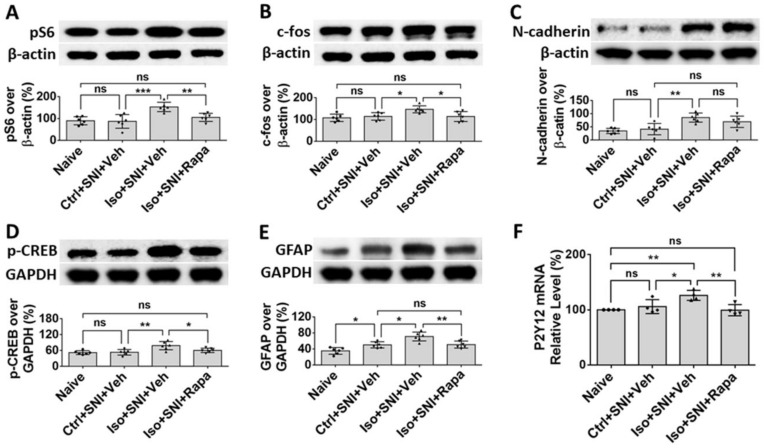
WB study shows alteration of neural activity markers in DSC. (**A**) The ratio of pS6 band intensity over β-actin in naïve and mice with SNI-produced chronic pain was at a same level. The amount in Iso+SNI+Veh group was dramatically higher than in Ctrl+SNI+Veh group and was attenuated by Rapa treatment close to naïve. (**B**) The intensity pattern of c-fos was similar to pS6. Numbers for naïve and Ctrl+SNI+Veh groups were identical. Iso+SNI+Veh enhanced, and Rapa reversed the reactive intensity. (**C**) Ctrl+SNI+Veh did not, but Iso+SNI+Veh did elevate molecule level of N-cadherin. Rapamycin restored this level to Ctrl+SNI+Veh animals. (**D**) The ratio of p-CREB over GAPDH, another standard marker, in naïve and Ctrl+SNI+Veh was almost same, but the amount in Iso+SNI+Veh was significantly higher. Rapa attenuated the level of p-CREB, and this number was near naïve. (**E**) The ratio of GFAP band intensity over GAPDH in Ctrl+SNI+Veh was elevated compared to naïve. This number was raised by Iso+SNI+Veh and a significant recovery resulted from Rapa treatment. (**F**) The mRNA of P2Y12 in DSC was examined with qPCR. Ctrl+SNI+Veh partially enhanced the relative level of P2Y12 than naive and Iso+SNI+Veh obviously increased P2Y12 than SNI only. This number was reversed by Rapa treatment close to naive. One-way ANOVA for all figures (*n* = 6 for A, B, C, D, E; *n* = 4 for F). ns: no significance; *: *p* < 0.05; **: *p* < 0.01; ***: *p* < 0.001.

**Figure 7 ijms-24-13760-f007:**
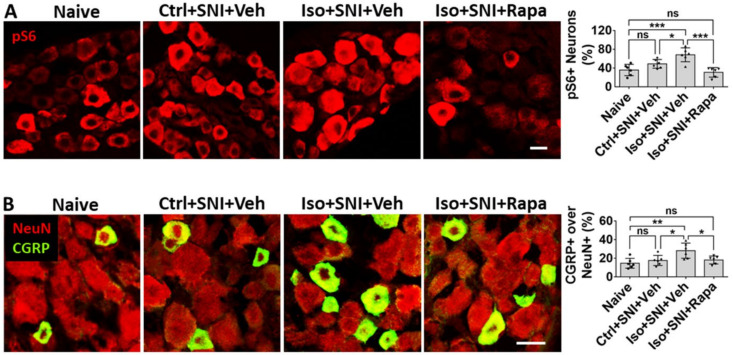
Effect of early isoflurane exposure on DRG neurons with IHC. (**A**) Effect of isoflurane and SNI on neuronal activity in DRG. In the chronic pain model (Ctrl+SNI+Veh), the number of pS6+ over all neurons was higher than in naive, but not statistically significant. Isoflurane obviously increased and rapamycin dramatically decreased this number, and it was even lower than naïve. (**B**) In naïve, CGRP was mainly expressed in small-sized (<25 µm) neurons. In Ctrl+SNI+Veh mice, the percentage of CGRP+ over total neurons was identical with naïve. In the Iso+SNI+Veh group, this number was increased and certain number of medium sized CGRP+ neurons (>25 µm) appeared. Rapa restored the percentage of CGRP+ neurons near naive. Bars = 25 µm. One-way ANOVA (*n* = 6). ns: no significance; *: *p* < 0.05; **: *p* < 0.01; ***: *p* < 0.001.

**Figure 8 ijms-24-13760-f008:**
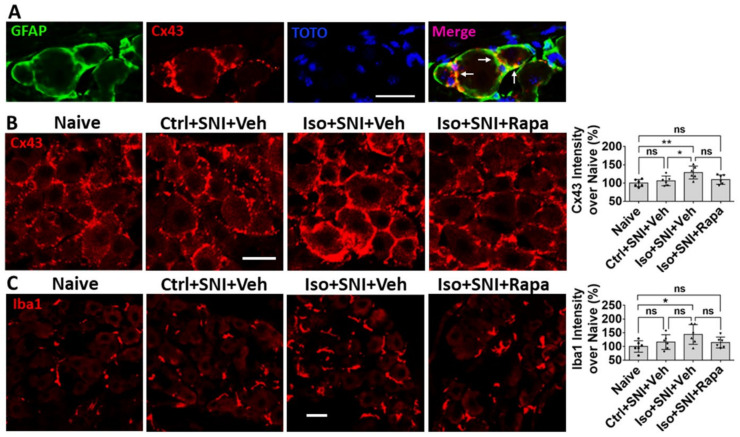
Effect of early isoflurane exposure on glial cells in DRG with IHC. (**A**) Connexin 43 (Cx43) is a gap junction protein in satellite glial cells (SGCs) and functions as neuronal coupling in DRG. The representative images show Cx43 is expressed in GFAP+ SGCs. Notice arrows indicate that Cx43 is predominantly located in the contacting region between neighbor SGCs. (**B**) Ctrl+SNI+Veh partially, but Iso+SNI+Veh significantly, increased Cx43 immunoreactivity. Rapa reduced Cx43 expression near naive. (**C**) The Iba1+ immunoreactivity in macrophage in DRG was measured. Ctrl+SNI+Veh did not, but Iso+SNI+Veh did, increase Iba1 fluorescence intensity than naive. Iso+SNI+Rapa restored Iba1 expression near naïve. Bars = 25 µm. One-way ANOVA (*n* = 6). ns: no significance; *: *p* < 0.05; **: *p* < 0.01.

**Figure 9 ijms-24-13760-f009:**
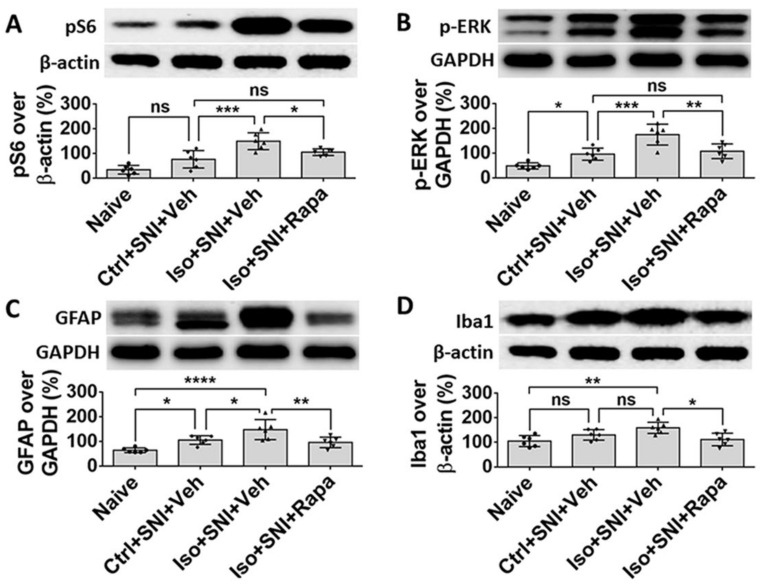
WB data show the effect of isoflurane exposure on mTOR expression and neural activity in DRG. The ratio of detected molecule density over standard markers, β-actin or GAPDH, was calculated. (**A**) Ctrl+SNI+Veh increased pS6 level to higher than naive, but the difference was not significant. Iso+SNI+Veh elevated this number by a large margin and Rapa reduced the pS6 intensity. (**B**) A neuronal activity marker, p-ERK, was used to detect neuronal activity. Ctrl+SNI+Veh increased p-ERK level to higher than naïve and Iso+SNI+Veh produced a huge increase. Rapa reduced the p-ERK react density. (**C**) GFAP was used to evaluate the level of SGCs. Ctrl+SNI+Veh increased GFAP level than naïve and Iso+SNI+Veh further upregulated the intensity than Ctrl+SNI+Veh. This number was reversed with Rapa treatment. (**D**) Ctrl+SNI+Veh did not, but Iso+SNI+Veh did, significantly increase Iba1 level to higher than naïve. Iso+SNI+Rapa downregulated iba1 reactivity close to naive. One-way ANOVA (*n* = 6). ns: no significance; *: *p* < 0.05; **: *p* < 0.01; ***: *p* < 0.001; ****: *p* < 0.0001.

## Data Availability

The datasets generated during the current study are available from the corresponding author upon reasonable request.

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
