# Peer review of "Effects of Early Exposure to Isoflurane on Susceptibility to Chronic Pain Are Mediated by Increased Neural Activity Due to Actions of the Mammalian Target of the Rapamycin Pathway"

_ijms, 2023, doi:10.3390/ijms241813760_

Round 1

Reviewer 1 Report

In the present paper the authors investigated the effects of anesthetics on the functioning of DRG neurons as well as glial cells. Overall, I think that this manuscript shows rich and valuable content, which is within the journal’s scope. Although the study is of principle interest to the neuroscience community, it covers several major experimental shortnesses that prohibit my recommendation for publication in the present form.

My comments:

Figure 6, 9 – please provide uncropped WB gels.

Line 88 – rapamycin was already abbreviated to rapa at line 88. The authors should consequently use this acronym throughout the manuscript.

Line 222 – From histological point of view there are only four kinds of tissues: epithelial, connective, muscular and nervous. Therefore, such terms as “DSC tissue” or “DRG tissue” are not justified.

Line 480 – please provide approval number of Ethical Committee.

Line 561 – the authors should explain why they focus on L4-L6 DRGs only and why they did not dissect the remaining L1-L3 DRGs.

Line 581 – the authors did not check the specificities of antibodies used. Pre-adsorption tests art lacking.

Line 589 - the authors should ensure that they use the term “expression” in relation to genes only.

Line 598 – immunohistochemistry is qualitative but not quantitative method. Because of rapid burning and fading effect any measurement of fluorescence intensity makes no sense.

Line 600 – how the authors measured the size of DRG neurons. What were criteria for size classification into small, medium and large?

Line 600 – How many researchers participated in counting. Were they blinded to the research group?

Line 606 – what was a lysis buffer?

Line 611 – what kind of blotting method was used (dry, semi-dry, wet)?

Line 634 - how the authors designed and synthesized primers.

Line 634 - Did the authors determine primers amplification efficiency?

Line 641 – Which two genes were used for reference purposes?

Line 642 - did the authors checked the normality assumption for ANOVA?

Author Response

Attached is our respond to reviewer 1.

Reviewer 2 Report

Thank you for permitting me to review this manuscript 

The authors consider general anesthesia as being only isoflurane 

isoflurane is only one type of anesthesia and the results should only be described for this agent only , the title should iclude isoflurane anesthesia

I do not believe the clue to these hypothesis is through retrospective data set  , it probably needs adequate prospective studies , further more the main target should be the prevention of chronic pain not exposure to anesthetics supposed to enhance the occurrence of chronic pain as the authors recognized it , therefore  further research should be rephrazed  

Author Response

The attached is our respond to reviewer 2.

(Please see the attachment)

Round 2

Reviewer 1 Report

Unfortunately, the manuscript is corrected superficially.

1. Still full images (uncropped) of WB gels are missing.

2. Size criteria for counting neurons are not included.

3. There is no any control procedure of antibodies used.

4. Information about observers number are not added.
